

# Detection of malicious consumer interest packet with dynamic threshold values

Adnan Mahmood Qureshi[1], Nadeem Anjum[1], Rao Naveed Bin Rais[2], Masood Ur-Rehman[3] and Amir Qayyum[1]

[1] Computer Science, Capital University of Science and Technology, Islamabad, Pakistan
[2] College of Engineering and IT, Ajman University, Ajman, United Arab Emirates
[3] James Watt School of Engineering, University of Glasgow, Glasgow, UK

Corresponding author
Nadeem Anjum,
nadeem.anjum@cust.edu.pk

## ABSTRACT

As a promising next-generation network architecture, named data networking (NDN) supports name-based routing and in-network caching to retrieve content in an efficient, fast, and reliable manner. Most of the studies on NDN have proposed innovative and efficient caching mechanisms and retrieval of content via efficient routing. However, very few studies have targeted addressing the vulnerabilities in NDN architecture, which a malicious node can exploit to perform a content poisoning attack (CPA). This potentially results in polluting the in-network caches, the routing of content, and consequently isolates the legitimate content in the network. In the past, several efforts have been made to propose the mitigation strategies for the content poisoning attack, but to the best of our knowledge, no specific work has been done to address an emerging attack-surface in NDN, which we call an interest flooding attack. Handling this attack-surface can potentially make content poisoning attack mitigation schemes more effective, secure, and robust. Hence, in this article, we propose the addition of a security mechanism in the CPA mitigation scheme that is, Name-Key Based Forwarding and Multipath Forwarding Based Inband Probe, in which we block the malicious face of compromised consumers by monitoring the Cache-Miss Ratio values and the Queue Capacity at the Edge Routers. The malicious face is blocked when the cache-miss ratio hits the threshold value, which is adjusted dynamically through monitoring the cache-miss ratio and queue capacity values. The experimental results show that we are successful in mitigating the vulnerability of the CPA mitigation scheme by detecting and blocking the flooding interface, at the cost of very little verification overhead at the NDN Routers.

## INTRODUCTION

Named Data Networking (NDN) is a well-known and well-researched architecture for the next generation of the Internet, based on a data-centric approach. While the legacy network is based on a host-centric system, the NDN architecture has changed the Internet's communication model altogether (*Jacobson et al., 2009*). It allows the distribution of data that can be acquired from any content router from the network.

A content provider can produce the data in advance and place it as auxiliary storage that can be accessed by any consumer anytime, even if the producer gets offline. A producer does not have to be online, and a consumer does not have to be connected to the producer to fetch the data; Instead, the consumer can acquire data through in-networking caches. While NDN increases content availability in the network via in-network caching, the integrity of content becomes critical, given NDN's nature (*Tarkoma, Ain & Visala, 2009*). Hence, NDN opens several security-related issues that are not relevant to the legacy network communication. It includes some new types of data integrity attacks where a malicious or compromised node provides a corrupted copy of the content. These issues are often ignored in NDN-related communication and caching mechanisms and are our main focus in the article.

One of the most critical attack vectors in NDN is the Content Poisoning Attack. The attacker compromises the Content Router (CR), and this compromised CR sends a reply to the legit request with totally bogus or corrupted content. This poisoned content pollutes the in-network caches of intermediate NDN routers and thus deprives the consumers of the requested content's legitimate copy. *Hu et al. (2018)* proposed a comprehensive scheme to mitigate the Content Poisoning Attack (CPA). A special interest packet is generated by the consumer, which contains the hash of the poisoned data. This article is all about the identification and mitigation of security flaws that can be exploited by the attacker during this CPA mitigation process.

The research problem lies in the CPA mitigation scheme proposed by *Hu et al. (2018)*. A consumer with malicious intent can flood the network with the Interest packet containing the hash digest of legit or un-poisoned data. This hash is stored in its exclude filter field. During CPA mitigation, this packet can flood the network, which will enable multipath forwarding and on-demand verification of hash at the router. This flooding attack can severely affect the throughput of the network or even cause a denial of service for other legitimate consumers. Therefore, it is essential to mitigate and add this additional security feature along with CPA mitigation (*Qureshi & Anjum, 2020*).

In this article, we proposed a scheme to detect the flooding attack generated by the compromised Consumer. A satisfaction test is performed to check if the excluded interest packet is non-existent in the cache or a legit packet. If the cache miss ratio (of the excluded interest packet) reaches the threshold value, it is considered an attack. A lightweight parameter is added to the Content Store data structure, which stores cache miss counter value. This value is compared with the specified threshold value. When the cache miss counter reaches near that threshold value, an event is raised that blocks the incoming malicious face. Also, in our scheme, we made the threshold value adaptable. At first, the initial threshold value is calculated by taking the total buffer size and divided it by the verification rate. The proposed idea is that when cache miss ratio avg crosses 50%, and queue capacity saturates, the threshold value is reduced to half. This process continues until the value is thrashed to one.

The articles's main contribution is the addition of a security feature that fills up the attack surface that can be exploited by the malicious consumer. Our contributions are:

- Adjustment of the threshold value dynamically by monitoring the cache-miss ratio value and queue capacity.
- Detection and mitigation of the flooding attack of special interest packets generated while mitigating the content poisoning attack.

Further, this article is organized into five sections; the second section emphasizes the literature review and related work. The third section is the proposed approach, and in the fourth section, experiments and results are highlighted along with the conclusion in the fifth section.

## RELATED WORK

Any network's primary goal is to share web content, including photographs, texts, and videos. Implementing security standards and goals such as confidentiality, integrity, and accessibility can ensure robust and flawless communication. Privacy guarantees that only the approved individual shall access the data. Integrity means that the receiver's received data must be similar to the one sent by the sender. Availability ensures that network infrastructure should be available for an authorized user whenever he needs the service (*Wein et al., 2007*).

*Kumar et al. (2019)* and *Hassanein & Zulkernine (2015)* explained some of the most common attacks within the existing TCP/IP model such as Denial of Service (DoS) attack, Distributed Denial of Service (DDoS) attack, eavesdropping (snooping), masquerading, TCP Replay Attack, Man in the Middle Attack, repudiation, and traffic analysis attack. These legacy attacks are not possible in NDN because of the absence of the host, but with the advent of this new architecture, some new attack surfaces have emerged which need to be addressed and it is an active research area.

### NDN's data-centric security and security issues in NDN

At the network layer of NDN, data-centric security is mandated via a digital signature on each data packet. A digital signature is added by the content provider (producer) to every data packet associating the data to the packet name when data is being generated. Authentication can be performed by the consumer on the data packet by verifying the signature using the Content Providers' public key. This authentication can be performed even if the data is retrieved from some other entity other than the content provider (*Ribeiro et al., 2016*).

*Zhang et al. (2018)* stated that If a content providers' public key is not distributed or the consumer has no information about this public key, in that case, the data producer places the signing key name into the specific field of the data packet. It is known as the KeyLocator field. A consumer can acquire a public key by following this field of KeyLocator and can retrieve it just like a normal data packet. *Kumar et al. (2019)* explained some of the most common attacks within the existing TCP/IP model such as

**Table 1 NDN attack types.**

| Attack types | Adversary | Victim | Compromised security goal | NDN element involved in attack |
|---|---|---|---|---|
| Flooding attack of interest packet | Consumer | Consumer/ Router/Producer | Availability | PIT |
| Cache pollution attack | Consumer | Consumer | Availability | CS |
| Privacy attack | Consumer | Consumer | Confidentiality | CS |
| Content poisoning attack | Producer or Router | Consumer/Router | Integrity/Availability | CS |

Denial of Service (DoS) attack, Distributed Denial of Service (DDoS) attack, eavesdropping (snooping), masquerading, TCP Replay Attack, Man in the Middle Attack, repudiation, and traffic analysis attack. In a modification attack, the attacker does not only compromise the confidentiality of the data by accessing it but also compromises the integrity of the data by trying to alter it. However, this attack is not possible in NDN, because each piece of data is signed by the publisher, which the consumer can verify. However, if the router itself is compromised and alters the data packet, then a corrupted data packet may be sent to the consumer. Consumers after receiving the publishers' public key can validate this corrupted data. In a masquerading attack, the attacker masks his identity and impersonates to be another person so he/she can acquire some useful information about that person. However, this attack is also not possible in NDN because every piece of data chunk is signed by the publisher using his/her private key. In a replay attack, the attacker performs Man in the Middle attack and tries to get a copy of the message from the sender, then after modifying the message and he/she sends it to the receiver. The recipient assumes that the actual sender has forwarded the message but in fact, it is the modified message from the attacker with malicious intent. This type of attack is also not possible in NDN because the interest packet is identified by the name and for the uniqueness of the namespace in the network, a nonce is used. When the same interest packet reaches the router (with the same name and nonce), the router assumes the packet is duplicate and it is replayed; it will, therefore, be purged from the PIT table. NDN, therefore, protects itself at the network layer level from the replay attack. In NDN architecture, some inherent security features protect us from some of the legacy security attacks by default but still there are some emerging security concerns in this new architecture that needs to be addressed. Security, privacy, and access control are the three major domains that need to be covered in NDN architecture. Several attacks are possible in NDN such as Content Poisoning attack, Content pollution attack, Naming Attack, and Denial of Service attack. In privacy, it can be classified into five categories such as content privacy, signature privacy, client privacy, name privacy, and cache privacy (*Wein et al., 2007*; *Ahlgren et al., 2012*; *Zhang et al., 2018*; *Hassanein & Zulkernine, 2015*). In access control, there is some mechanism that needs to be addressed are content encryption, content attributes, clients' identity, and authorized sessions.

## Attack types in NDN

In NDN there are four main types of security threats that are briefly discussed in the coming sections and the attack effects on the security goals are mentioned in Table 1 (*Kumar et al., 2019*; *Tourani et al., 2018*).

### Foodig attack of interest packet

*Benmoussa et al. (2020*, *2019)* explained in details the effects of an interest flooding attack, in which, an attacker can deplete the network resources by flooding the network with a large number of interest packets batches. PIT, network bandwidth, and producer resources availability to the legit users will be compromised with this attack. This attack consumes NDN resources that restrict legitimate users from accessing them.

### Cache pollution attack

*Wang et al. (2017)* discussed the anatomy of cache pollution attack, the attacker attempts to fill the cache with unwanted content in the NDN router by demanding the data packets which are unpopular and not in demand. As a result, the NDN routers' impact ratio decreases. Therefore, the cache hit ratio of the interest packet of the legitimate user will thrash. This will increase the latency and reduce the throughput of the network.

### Cache privacy attack

During an assault on cache privacy, the attacker wants to figure out whether or not the sensitive data has been accessed recently. A newly accessed item lies in the routers' cache and the requester gets a quick response to these types of data. The intruder compiles a list of content that is vulnerable to privacy and asks them one by one to know whether it is cached or not by noticing the delay in retrieving the content. If the content is identified, the attacker can conclude that a user or a group of users has recently accessed the content. The adversary will know the user's access pattern using this technique. The content type that is accessed and other related information will also be vulnerable to privacy.

### Content poisoning attack

One of the most crucial attack vectors in NDN is the Content Poisoning Attack. In CPA, the attacker compromises the router, and this malicious router sends a reply to the legitimate request with totally bogus or corrupted content. The contents of intermediate routers that are involved in NDN communication are stored in CS. This poisoned content spreads when other legitimate consumers request the same content. Content in NDN is of three types, that is, legit contents, fake or poisonous contents, and corrupted contents. A valid signature of valid content is generated through the private key of a legit publisher. Similarly, a valid signature of fake content can also be generated with any private key that is not associated with the publisher's namespace. Whereas the corrupted content does not have a valid signature *Ullah et al. (2020)*. In a Content Poisoning Attack, an attacker takes over a router and replies to incoming interests with corrupted content. *Wu et al. (2016)* explained that if a consumer requests this corrupted content, it will spread this malicious content on intermediate routers' content stores. It will result in the spreading of

this poisonous content all over the network. This verification is usually performed by Consumers who use the content verification process using the content's signature. In NDN, every router can verify the arriving contents on its own, but this verification at line speed takes resources, and it is impractical.

*Gasti et al. (2012)* described two ways through which a content poisoning attack can be carried out. The first way is that the attacker compromises the routers, spreading the poisoned content while satisfying the requested interest packets. The second way is that poisoned content is distributed via compromised publishers. Compromised publishers can anticipate the Data that will be in high demand, for example, highlight a famous football match, and create malicious content. So in this way, a compromised producer or router can reply with a malicious data packet against a legitimate interest packet.

### CPA detection and mitigation approaches

Content Poisoning Attack can be detected and mitigated through two major approaches, Collaborative Signature Verification, and the Consumer Dependent approach. The former method is those in which NDN routers collaborate to verify the content's signature. The latter method uses extra fields in the Interest and Data packets or uses clients' feedback.

### Mitigation of CPA using consumer dependent approach

As per NDN specification, a consumer verifies all the signatures of the requested data packets. So a feedback-based approach is used to verify the content at the router (*Gasti et al., 2012*). This approach is the extended version of the NVF technique, as discussed in the previous section. However, this approach has some new challenges, such as there is no trust relationship between the router and the consumers. Consumers can also be compromised, and in this way, false feedback can consume network resources. *Ghali, Tsudik & Uzun (2014a)* proposed a technique for content ranking calculation and stored in the exclude field of the interest packet, and the range of the values are between 0 and 1. New content is ranked 1, which gets downgraded if rated by consumers and included in the excluded field of the consumer. This approach is somewhat similar to the technique mentioned in *Gasti et al. (2012)*, so it has the same limitations. *Ghali, Tsudik & Uzun (2014b)* highlighted some of the NDN architecture vulnerabilities, such as the PPKD field and name's digest are not the essential components of the Interest packet. Also, no such trust model is adopted unanimously by the consumer's applications to fetch the content's hash securely. Based on these vulnerabilities, a technique is proposed, which enables an IKB rule to ensure trust. According to this rule, the Interest packet must include the producer's (content publisher's) public key. It is also implied that producers should also have the public key in the Data Packets' KeyLocator field. Its implication on the router is that it should calculate the hash of the public key of the content received and compare it with the PPKD field against its PIT entry. Upon mismatch, the content is discarded but otherwise verified. Upon successful verification, content is forwarded and stored in the content store of that particular router. *Yue, Li & Pang (2019)* stated that IKB

implication for consumers is that it has to acquire and validate the content provider's public key before initiating the Interest packet for that specific data packet. Trust model can be acquired using three approaches: public keys of the content provider should be installed in the client application, the second one is the universal key name service, and the third one is global search-based service. Also, to reduce the core routers' workload, the author has proposed that an Interest Key Binding check on the data packet should be performed at the edge routers. In contrast, core routers should perform this check probabilistically. The cons of this approach are that it is assumed that verifying the router is trusted, but it can verify the bogus IKB to be correct if it is malicious. So this scheme lacks scalability and has overhead.

*DiBenedetto & Papadopoulos (2016)* proposed an approach in which consumers, upon verification failure, send a report packet, which will act as feedback to the other entities of the NDN Network. When consumers detect a poisoned content, a special interest packet is generated by the network stack, and the information regarding the poisoned content is stored in this special report packet. When the router receives this special interest packet, it acts as one of the two proposed mitigation options that the author proposed. One is Immediate Failover, and the second one is Probe First. In the first approach, the malicious face is marked with a low priority value for the future. And in the probe first technique, the node, upon receiving the special interest packet known as report packet, stops forwarding the interest packets of the namespaces on which the attack is underway. Also, that particular node informs their next-hop routers about this malicious namespace.

*Nguyen et al. (2017)* explained three vulnerabilities in NDN architecture; the first one is unregistered remote provider, then multicast forwarding and the last one is the best route forwarding. The first vulnerability is that the interest packet can be satisfied with any data packet received from any of the faces. Therefore, a malicious producer can induce malicious content and satisfy it before it gets satisfied by the legit producer. In NDN, faces are registered in the FIB table's corresponding values, so while doing multicast forwarding, the interest packet is forwarded to all these faces. So, the malicious producers can satisfy the interest packet with its malicious content. A router ignores a similar interest packet in the best route forwarding with the same name and selectors but different nonce when received during the suppression interval of retransmission. The interest received after this interval shall be transferred via the next lowest possible cost; thus, an interest packet can be satisfied with a malicious producer's poisoned contents.

*Hu et al. (2018)* proposed a comprehensive system to mitigate CPA, and this article is all about identifying security flaws and proposing a mitigation strategy to address this flaw in this system. In the following sections, this base system is elaborated in detail. This system is comprised of three phases. First is the route building phase, then there is a normal content retrieval phase, and the last one is the recovery phase in chase content poisoning. It is required that NDN routers should enable name-key-based forwarding to forward interest towards registered content sources, and to specify legitimate content sources, every route advertisement should be authenticated with a trust management system. If content poisoning occurs on intermediate routers, then a mechanism of

**Table 2 CPA detection and mitigation.**

| References | NDN node | Detection | Mitigation | Overhead |
|---|---|---|---|---|
| *Gasti et al. (2012)* | Consumer, Router | Signature, PPKD | SSCIC, DSCIC | Verification of random signatures |
| *Ghali, Tsudik & Uzun (2014a)* | Consumer | Signature | Content ranking | Content ranking calculation |
| *Ghali, Tsudik & Uzun (2014b)* | Router | PPKD and signature | Interest key binding | Signature verification |
| *Nam, Kim & Yeom (2015)* | Router | Signature | SLRU extension | Signature verification |
| *Kim et al. (2015)* | Router | Signature | SLRU extension | Signature verification |
| *Wu et al. (2016)* | Consumer, router | Signature | Reputation based forwarding | Signature verification |
| *Kim et al. (2017)* | Router | Signature in case of cache-hit | SLRU extension | Signature verification |
| *DiBenedetto & Papadopoulos (2016)* | Consumer, router | Signature | Modified forwarding strategy | Complete Bogus packet in reissued interest packet |
| *Hu et al. (2018)* | Consumer, router | PPKD and signature | Name-key based forwarding and multipath forwarding based inband probe | Signature verification (Hash matching is fast due to PPKD entry) |

"Multipath Forwarding based Inband Probe" is performed. In this approach, an interest packet with an exclude filter (poisoned content) is reissued and forwarded via alternate paths. When this packet reaches a particular router, it enables cached contents' on-demand signature verification. Verification of cached content is performed between the malicious payload included in the interest packets' exclude filter or in the Data Packet that is returned and gets matched with the reissued interest packet. There are two benefits of this approach; first, with multipath forwarding, there is a great chance that consumers will acquire the legitimate content while legitimate content can be restored on the intermediate router via alternative forwarding options. This way, poisoned contents will be purged, and for future requests, legitimate contents will be returned from the routers' cache. Thus it will increase the overall throughput of the network.

### Comparisons of CPA mitigation approaches

Table 2 is a summarized view of the CPA Mitigation approaches, as discussed in previous sections:

Based on the analysis of existing techniques and work to detect and mitigate CPA, there is still a need to sort out some challenges while developing a CPA mitigation strategy (*Hu et al., 2018*). Energy management in routers is an important issue. *Gao, Wang & He (2018)* evaluated that CPA and caching issues can consume a considerable amount of routers' energy, which can add instability to the whole system. In *Hu et al. (2018)* have implemented a robust and efficient mechanism to mitigate the CPA. In Table 3, we have identified vulnerabilities in content poisoning attack mitigation schemes discussed in

**Table 3 Vulnerabilities in CPA mitigation schemes (compromised consumers can flood routers).**

| References | Checked by | Proposed solution | Energy efficient | Security features |
|---|---|---|---|---|
| *Gasti et al. (2012)* | Consumer and router | SSCIC & DSCIC | Yes | Cannot detect corrupted content |
| *Ghali, Tsudik & Uzun (2014a)* | Consumer | Content ranking algorithm | No—Overhead of calculating the content ratings | Do not handle malicious consumer in case it reports false content rating |
| *DiBenedetto & Papadopoulos (2016)* | First consumer and then router | Modifying forwarding strategy | No—uses complete bogus packet in report | Only handles the malicious consumer identity but do not handle the corrupted data |
| *Hu et al. (2018)* | First consumer and then router | Name-key based forwarding and multipath forwarding based inband probe | Yes—only use a PPKD extra field and use bogus/corrupted data hash in excluded filter field of interest packet | Can prevent poisoning of content by generating special interest packets |

previous sections of this article. In the following section, we explore how these vulnerabilities can be exploited and a mitigation strategy is proposed, which is the main research area of this article.

# PROPOSED APPROACH

## Introduction

The name-key based forwarding and Multipath forwarding based Inband probe is a very comprehensive scheme for mitigation of the CPA. It fills most of the attack surfaces regarding the Content Poisoning Attack. However, with the advent of the NDN architecture's structural changes, it has induced a new attack vector that can be exploited by the adversary. With this attack, the whole system can collapse. So it is very crucial to highlight this aspect. One of the important attack vectors that have emerged with this technique is the flooding of the reissued Interest Packet containing the excluded filter field. It is the leading research contribution of this article. A consumer with malicious intent can flood the network with interest containing the hash digest of legit or unpoisoned data in its exclude field, which can flood the network and enable multipath. It can harm the throughput of the network or even can cause DDoS. Based on the research gap mention in the previous section, this article has formulated the following research questions: What will be the mechanism to detect the attack initiated by consumers with malicious intent? What will be the parameters that will mitigate the malicious consumers' reissued interest packet flooding attack? So it's essential to mitigate and add this additional security feature in this CPA Mitigation technique.

### *Detection of malicious consumer interest packet with excluded filter field*

During the CPA, the reissued Interest Packet by the consumer stores the hash of the poisoned data in the excluded filter field but a compromised consumer can also store the hash of and un-poisoned data has in the same field. Consequently, It will result in a cache miss. The on-demand signature verification at the router will also be enabled during this process, consuming a lot of router processing power. When a consumer with malicious intent bombards these excluded Interest packets, although they get discarded at the

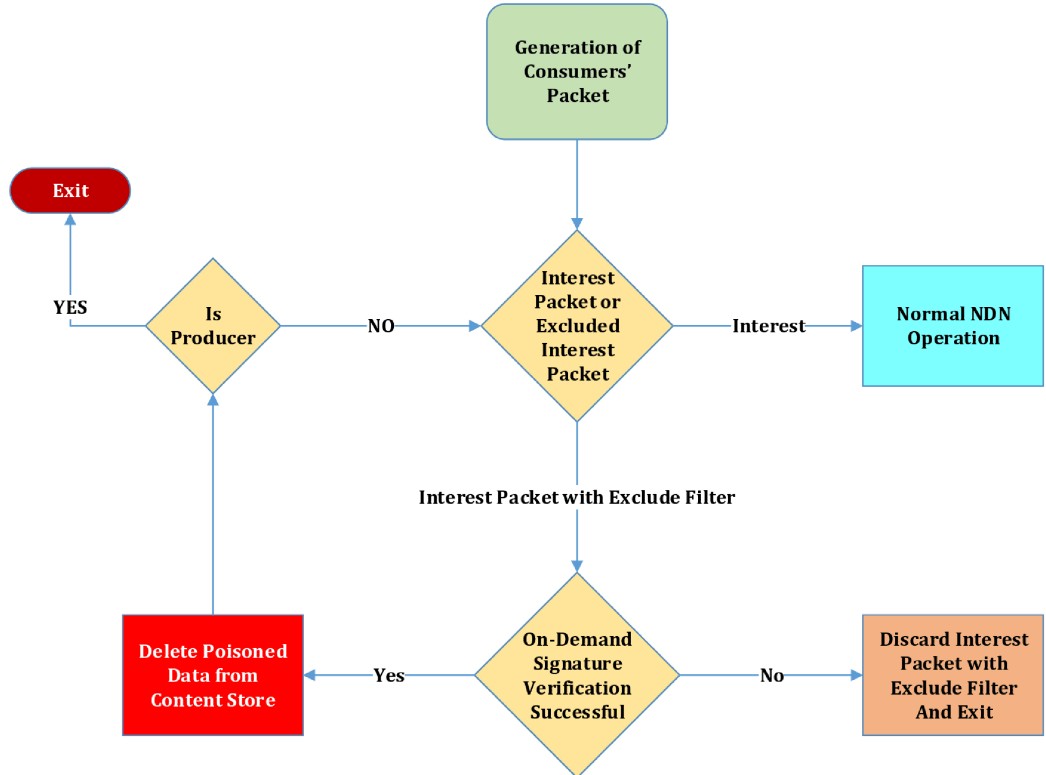

**Figure 1  Detection of flooding attack during content poisoning attack mitigation.**

next router upon verification, it will drastically increase the router's processing overhead. Other legitimate consumers will face a denial of service from this router. This attack vector should be taken into account, and a mitigation strategy should be devised for such attacks. This way, the process of CPA mitigation will be severely affected. The block diagram of the flooding scenario is elaborated in Fig. 1. The block diagram depicts the scenario of the flooding attack of Interest Packet with excluded filter. The first block shows that the consumer generates the normal Interest packet. Then a decision is taken in the next block that whether it is a normal Interest packet or an Interest packet with an excluded filter field. In case it is a normal Interest packet, it is directed towards normal NDN operations; otherwise, it is passed to the next module of On-Demand Signature Verification. Here signature verification is performed against PPKD in the Content Store. If validation fails, this packet is discarded; otherwise, it gets purged from the router's CS. In the case of poisoned Data is found in the CS, the normal process is initiated, and content poisoning mitigation will commence. When a consumer is compromised, and it starts flooding the NDN Network with the excluded filter enabled Interest packet, it will trigger On-Demand Signature verification for each bogus packet, and the next NDN router will get saturated. The queue will be occupied, and after a while, there will be less space for the legitimate excluded filter Interest packet. It will hamper the CPA Mitigation mechanism badly. So this scenario is considered an attack and needs mitigation.

### Mitigation of flooding attack

In this article, a reactive approach is proposed to mitigate this attack. A virtual queue is utilized in NDN Routers for incoming reissued Interest packets from the consumers. FIFO (FCFS) queue is shared among all the incoming faces for reissued interest packets. It is a temporary place holder for these packets until they get verified. The allotted memory for the transmitting packets should be different from the one used for caching. If the same CS is used to transmit packets and data chunk, then the CS will be congested with the data chunks that are waiting to be satisfied by the pending Interest packets. To prevent the Malicious consumer from sending a fake "excluded Interest packet," a satisfaction test is performed to check if the excluded interest packet is non-existent in the cache or a legit packet in the cache. In case a cache miss (of the excluded interest packet) occurs, and the ratio reaches near the threshold value, that is, it is set by the operator, it is considered an attack. On-demand verification at the router is not enabled unless there is a cache hit of the excluded interest packet; this will reduce the overhead of content verification at each data packet's router. However, in case of a cache miss, this excluded interest packet is discarded. Still, if a consumer with malicious intent floods the edge router with the fake interest packet with the excludes filter, it will degrade that particular edge router's performance. The NDN-router service manager at the NDN Router, especially at the edge of the network in the consumer domain, maintains the stats and looks at them. The router will drop the future reissued interests coming from this face with the excluded data packet as it is considered a malicious consumer upon hitting the threshold value. It will be done temporarily and delisted at the discretion of the network operator. A new lightweight parameter is added in the CS Data Structure to retain the cache miss counter of invalid reissued Interest packet with excluded filter field. This value is compared with the threshold value. The Block diagram can show the birds-eye view of this proposed mechanism in Fig. 2.

We have introduced a block of the proposed approach. The reissued Interest packet upon several caches misses, and hitting the specified threshold value will trigger an event and blocks this malicious face. On the next iteration, this reissued packet from the malicious consumer face will be blocked. In Algorithm 1 (Fig. 3) PPKD, ContentName, nonce, incoming face, excluded filter field value, Threshold value, and cache miss counter value is passed as an argument. At statement 1, the hash comparison is performed and if the result is a cache miss then the cache miss counter value gets incremented. If that value reaches the threshold value then the event to block that specific malicious incoming face gets triggered. In case the result is a cache hit then the normal NDN communication process will commence.

### Dynamic threshold value

This approach helps the Network Operators set up the threshold value automatically during the special interest packet flooding attack by a malicious consumer. This approach aims to select the threshold value in an automated fashion based on the statistical monitoring of buffer capacity and cache miss ratio. In this approach, a Network Management software continuously monitors the cache miss ratio and buffer capacity

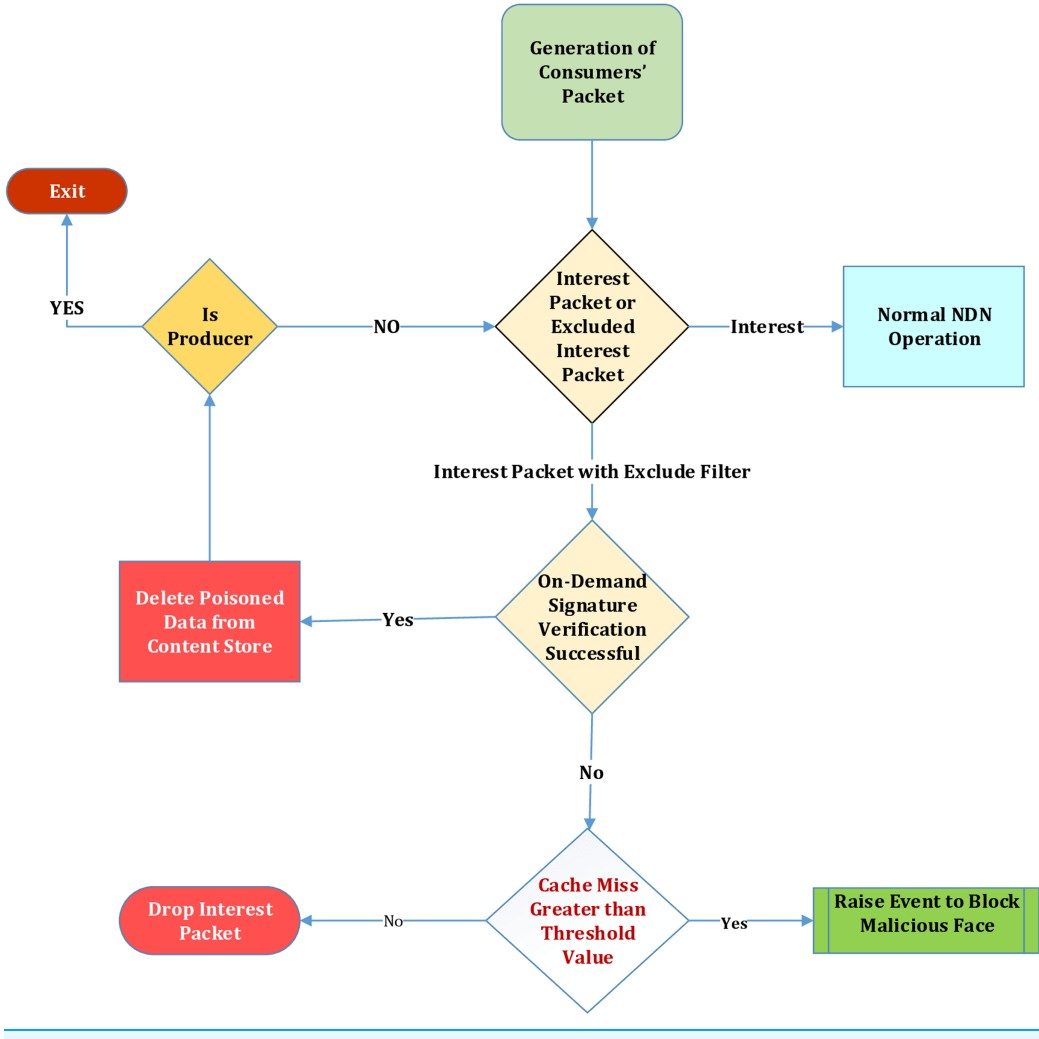

**Figure 2 Mitigation of flooding attack.**

when a special interest packet is initiated. When the cache miss ratio average over a while results in a buffer overflow, the threshold value is thrashed to half. This process continues unless the threshold value becomes 1. This mechanism is elaborated in Algorithm 2 (Fig. 4). At this stage, the incoming face causing the flooding attack will get blocked till the particular timeout.

$$InitTH = QueueSize/VerificationRate \qquad (1)$$

$$Cache\_Miss\_Ratio = CM/(CM + CH) \qquad (2)$$

Network Management Software will continuously monitor the Cache_Miss_Ratio and Buffer Size of the queue.

### Benefits of dynamic threshold values over static threshold values

The mitigation of flooding attack of special interest packets works on two approaches, first one uses the static threshold values which is set by network operators during the router

```
Algorithm 1: Detection & Mitigation of Flooding Attack of Reissued Interest Packet
   Require:    REISSUED_INTEREST_PACKET    (Content_Name;    PPKD;    nonce;
   IncomingInterface; excluded, THRESHHOLD, CACHE_MISS_COUNTER)
           //RESULT is Enumerated Data having values CACHE_HIT & CACHE_MISS
   1:   RESULT ← HashComparison (PPKD_HASH_ExcludeFilter; PPKD_CS)
   2:   if RESULT = CACHE_MISS then
   3:         CACHE_MISS_COUNTER += 1
   4:         if CACHE_MISS_COUNTER = THRESHOLD then
   5:               RAISE_EVENT (FLOODING_ATTACK_MITIGATION)
   6:               RAISE_EVENT (INCOMING-INTERFACE-POISONED-FLAG)
   7:               PIT.DROP (Interest)
   6:               EXIT
   7:         end if
   8:   else if RESULT=CACHE_HIT
   9:   ContentStoreEntry ← FindInContentStore (Content_Name; PPKD)
   10:  if ContentStoreEntry > 0 then
   11:        if INTEREST.HasExcludedData ( ) = False then
   12:              Return ContentInCS
   13:        else
   14:              if HasExcludedData (DATA) = True then
   15:                    Raise EVENT (FibEntry) //Alert: FibEntry
   16:              end if
   17:              if IsPoisoned (DATA) = True then
   18:                    Purge (this.DATA)
   19:                    NextHop.Cost ++
   20:              end if
   21:        end if
   22:  else
   23:        PITCount ← FindInPIT (Content_Name; PPKD; ExcludedFilter)
   24:        if PITCount is > 0 and nonce.IsUnique = TRUE then
   25:              ifaces.Add (Incoming_Interfaces)
   26:        else
   27:              FIBCount ← FindInFIB (Content_Name; PPKD)
   28:              if FIBCount is > 0 then
   29:                    if NextHop hop= 1 then
   30:                          ForwardInterest (hop)
   31:                    else
   32:                          foreach hop & flag = null
   33:                          Malicious iface do
   34:                                ForwardInterest (hop)
                                  \\ Multiple Paths
   35:                          end for
   36:                    end if
   37:              end if
   38:        end if
   39:  end if
```

**Figure 3** **Detection and mitigation of flooding attack of reissued interest packet.**

initial configuration. The second approach is the dynamic approach, in which the threshold value is adjusted adaptively by monitoring the Queue size and Cache Miss Ratio value.

# EXPERIMENTAL RESULTS

## Simulation environment

For proof of concept and to run this scenario, a custom-built NDN Simulator is developed in C# language in Visual Studio 2019. The network parameters used in simulation scenarios are mentioned in Table 4.

```
Algorithm 2: Automated Threshold Value

Require: SET_THRESHOLD_VALUE (Total_Queue_Size; IncomingInterface;
PktVerificationRate, Cache_Miss_Value, Cache_Hit_Value)

//Initialize Threshold Value

1: THRESHOLD_VALUE = Total_Queue_Size / PktVerificationRate

2: Current_Queue_Size = FUNCTION_Get_Queue_Occupation( )

3: WHILE ( TRUE )

4: IF (THRESHOLD_VALUE<=1) THEN RETURN 1

5: CACHE_MISS_RATIO = CACHE_MISS_VALUE / (CACHE_MISS_VALUE + CACHE_HIT_VALUE)

6: IF ( Current_Queue_Size / Total_Queue_Size >= 1) THEN

7:      IF (CACHE_MISS_RATIO > 50) THEN

8:           THRESHOLD_VALUE = THRESHOLD_VALUE * 1/2

9:      END IF

10: END IF

11: RETURN THRESHOLD_VALUE

12: END WHILE
```

**Figure 4  Dynamic threshold value.**

**Table 4  Simulation parameters.**

| Parameter | Default value |
| --- | --- |
| Request rate | 100 Interests/second/consumer (interest with exclude parameter) |
| Max queue length | 500 (Experiment 1 and Experiment 2) 1,000 (Experiment 3 and Experiment 4) |
| Verification of interest packet | 25 Interest/second |
| No. of malicious consumers | 1 (Experiment 1 and Experiment 2) 2 (Experiment 3 and Experiment 4) |
| Threshold value | x |

## Network topology

### Scenario 1: one malicious consumer

In scenario 1, our simulations' network topology consists of two routes from the consumer to the producer. Two paths routes that are used in this scenario are 0-1-2-4-6-7-8 and 0-1-3-5-7-8; these paths are between the consumer and a producer (*Spring et al., 2004*). In this scenario, it is evident that consumers with malicious intent can flood the network with unwanted interest packets with excluded fields occupied by the non-malicious or legit payload. If not mitigated at the edge router, all the routers will enable the on-demand verification, and this way, router performance will degrade with time. This problem can be mitigated by enabling a mechanism at edge routers of NDN and setting a threshold value that if it hits this value, block that interface through which these malicious excluded interest packets are coming. This way, the rest of the network will be safe from acquiring this malicious packet from consumers, and ultimately the performance of the intermediate routers will not be degraded. So to handle this issue Network Manager at NDN Edge

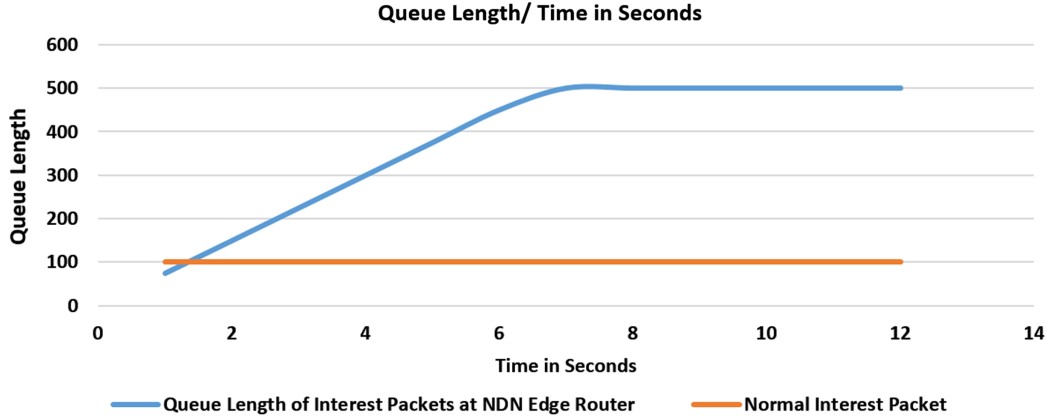

**Figure 5 Flooding attack with no threshold value and with one malicious consumer.**

Router enables this mechanism in which malicious interest packet with exclude field is dropped in case of a cache miss, and upon hitting the threshold value, the interface from which these excluded interest packets are received is blocked and added to the delist data structure. The timeout to get out of this delist data structure is at the desecration of the network operator.

### Scenario 2: two malicious consumers

In scenario 2, our simulations' network topology consists of two routes from two consumers (i.e., Consumer 1 and Consumer 2) of the same domain to the producer via Router 8 (edge router). The routes that are used in this scenario is 0-1-2-4-6-7-8 and 0-1-3-5-7-8; these paths are between the consumer and a producer (*Spring et al., 2004*). The main thing to note in this scenario is that Consumer 1 and Consumer 2 are in the same domain. Router 8, the virtual queue for Incoming Reissued Interests, is shared between these two consumers. The Queuing mechanism used in this scenario is FIFO. There are two consumers with malicious intent in this scenario and can flood the network with unwanted interest packets with excluded fields occupied by the non-malicious or legit payload. If not mitigated at the edge router, the virtual queues will be fully occupied for the legit reissued interest packet, and consequently, packets will drop. This problem can be mitigated by enabling a mechanism at edge routers of NDN and setting a threshold value that if it hits this value, block that interface through which these malicious excluded interest packets arrive. This way, the rest of the network will be safe from acquiring this malicious packet from consumers, and ultimately the performance of the intermediate routers will not be degraded.

## Experiments and result
### Experiment 1 (Scenario 1): with no threshold values
In this experiment as shown in Fig. 5, we have calculated the cache miss ratio of the interest packet containing the exclude filter and compared it with the Queue Length. Upon flooding the router with a fake interest packet, the verification process takes time, and

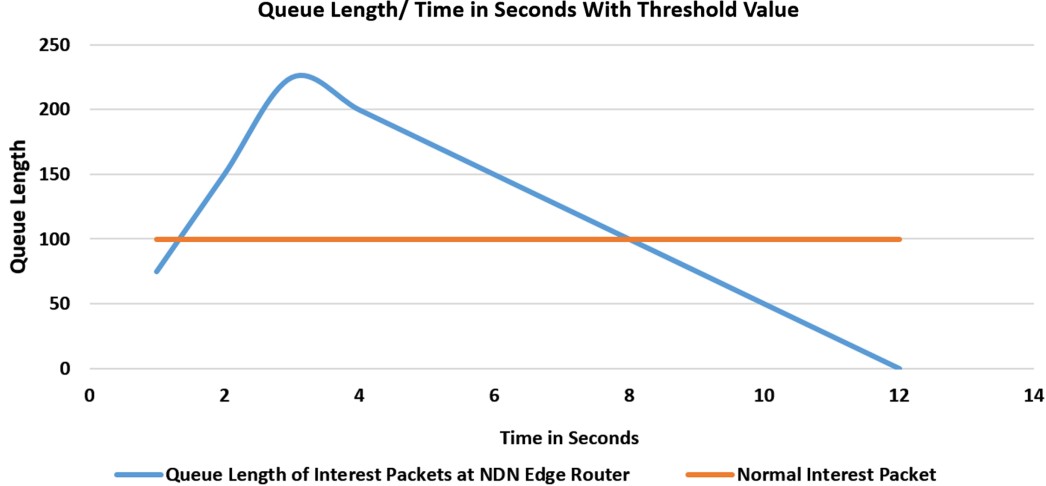

**Figure 6 Flooding attack with threshold value and with one malicious consumer.**

meanwhile, the queue of interest packets will start increasing. After every second, 25% fake packet will drop, and 75% will be added to the queue. Initially, no threshold value is set. After some time, congestion at the router's incoming interest packet queue will occur, resulting in a drop of other future packets at this router.

### Experiment 2 (Scenario 1): with threshold values

In the second experiment as shown in Fig. 6, our proposed scheme is enabled at the edge routers in Network Management software. After several cache misses and upon hitting the threshold value to 3 according to the simulation settings, it will block the incoming face of the consumer, and further, no more interest packets will be received from this malicious consumer face. After hitting the specified threshold value, the face is blocked and fake packets begin to drop from the queue. At 12 s the queue will be empty and the router is no more saturated.

### Experiment 3 (Scenario 2): with no threshold values

In the third experiment as shown in Fig. 7, Consumer 1 starts flooding the network with fake interest packets with the excluded filter; the queue will begin to saturate as the verification rate is slow as compared to the flooding rate. In the 6th second, Consumer 2 also starts to flood the network; consequently, the queue begins to saturate linearly.

Initially, no threshold value is set, and at the 9th-second congestion at the router's incoming interest packet queue will occur which will result in a drop of other future packets at this router.

### Experiment 4 (Scenario 2): with threshold values

In the fourth experiment as shown in Fig. 8, our proposed scheme is enabled at the edge routers in Network Management software. Upon cache miss threshold value reaches 3, it will block the incoming face of Consumer 1 after three failed verification at 4th second. Further, no interest will be received from this malicious consumer face. At 6th second,

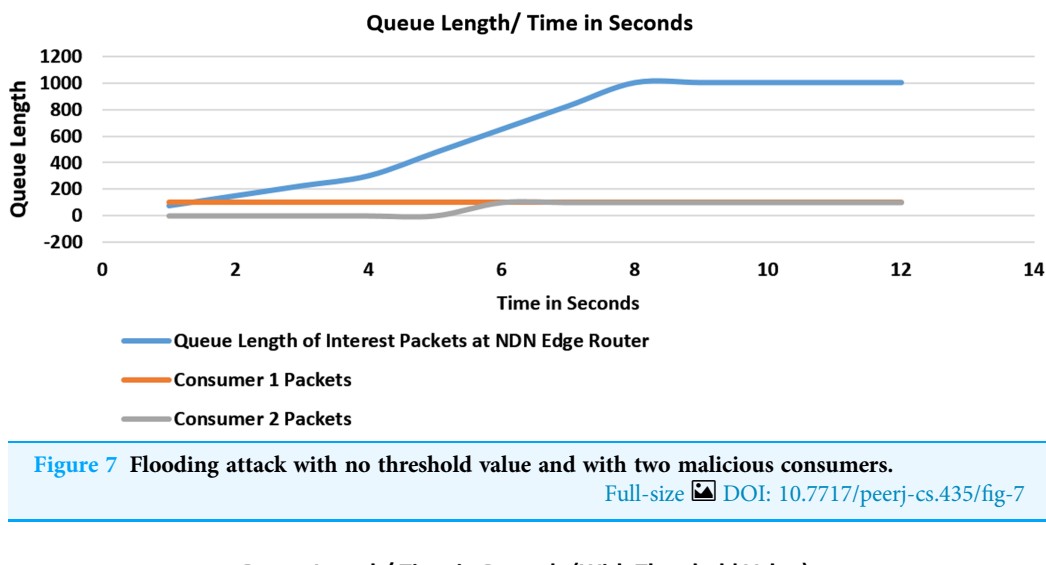

**Figure 7 Flooding attack with no threshold value and with two malicious consumers.**

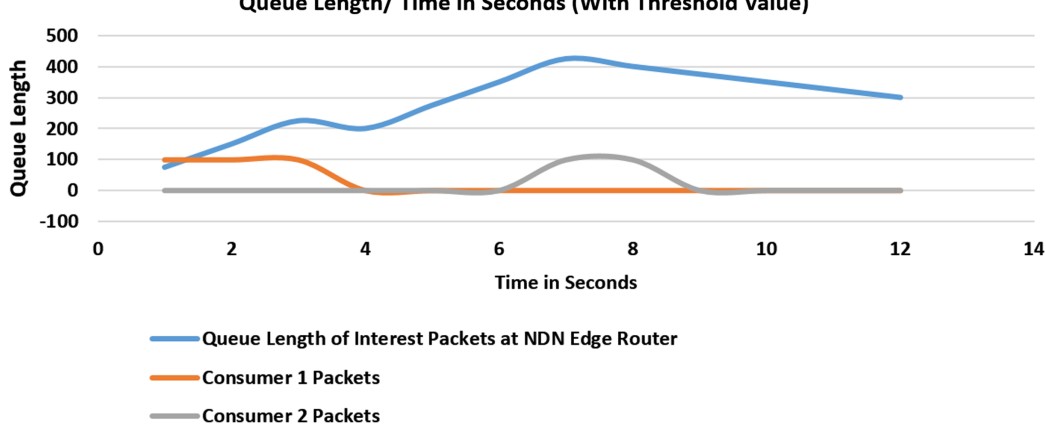

**Figure 8 Flooding attack with threshold value and with two malicious consumers.**

Malicious Consumer 2 starts to saturate the queue which will, and similarly, after three failed attempts, this face gets blocked as well, and queues start to thrashed after both of the malicious consumer faces are blocked.

## Dynamic threshold value

It is evident in the experiment that with the increase in the Cache_Miss_Ratio, the Queue size will increase because the flooding rate is greater than the verification rate. Also, Cache_Miss has a penality on the processor of the router, which can increase the processing overhead. In the graph (Fig. 9), the initial threshold value is set to buffer size divided by the packet verification rate. At 7th second the queue is filled up to 100%. At this stage, the new packets will start to drop. Here, the system should act prudently and reduce the threshold value to half of the current value, and if flooding continues threshold value is reduced to half as shown in Fig. 10, and so on till the value is reduced to 1. At this stage, the incoming face is blocked as it is considered as an attack. The queue will not be saturated, and memory will be available for other interest packets to get processed.

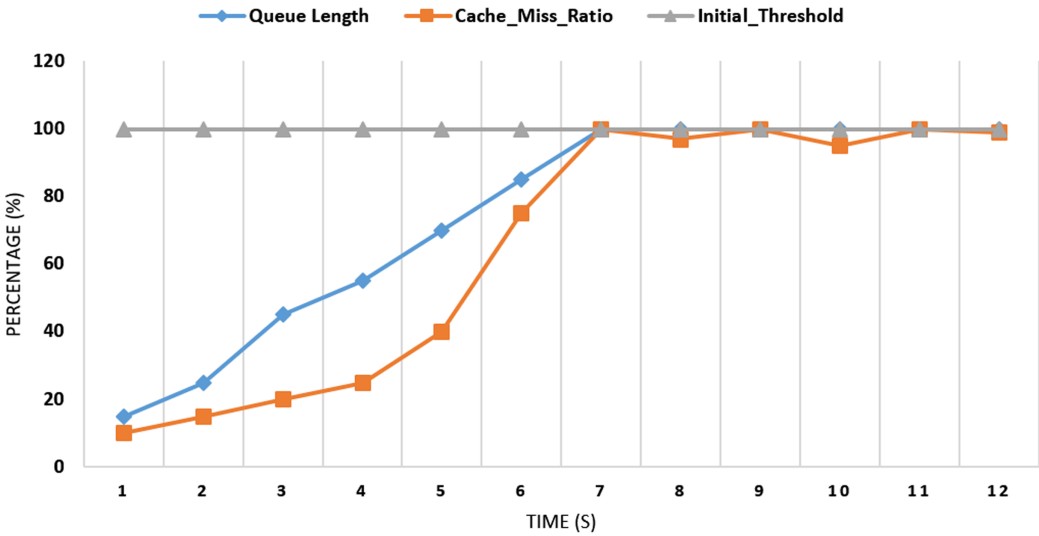

**Figure 9 Special interest packet flooding without dynamic threshold value.**

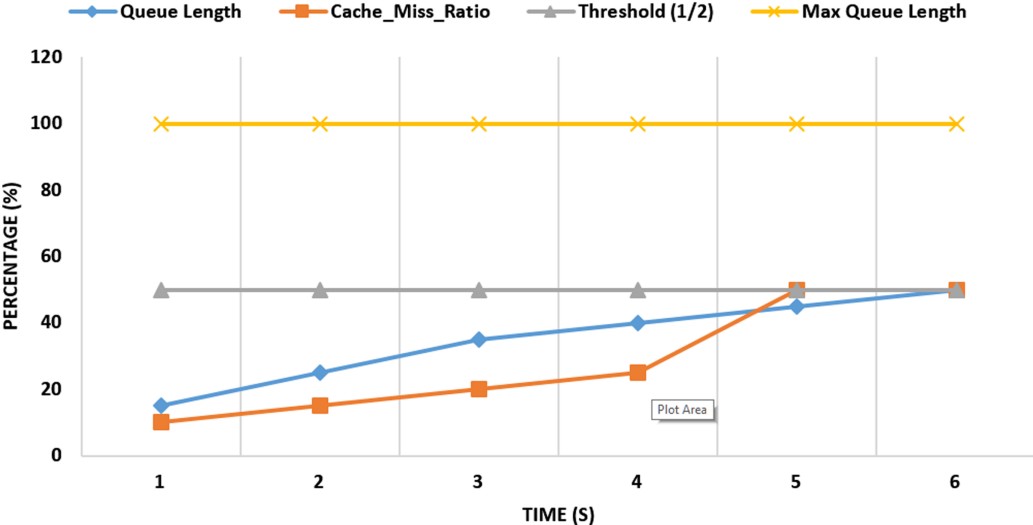

**Figure 10 Special interest packet flooding with dynamic threshold value.**

If the flooding attack continues, we will multiplicatively decrease the threshold value to another half. This mechanism will continue against that particular flooding malicious face until the threshold value reaches 1. At this stage, that particular face will be blocked and considered as a malicious face. The face will be blocked until the timeout, whose value will be at the network operator's discretion.

**Table 5 Simulation parameters for effectiveness of proposed approach.**

| Parameter | Default value |
|---|---|
| Request rate | 100 Interests/second/consumer (special interest packets) |
| Interest packet max queue length | 500 |
| Verification of interest packet | 25 Interest/second |
| Number of malicious special interest packets | 2,000 pkts |
| Number of normal special interest packets | 1,000 pkts |
| Number of malicious consumers | 1 |
| Threshold value | x |

## Effectiveness and accuracy of proposed solution by comparing the throughput of the normal special interest packets

The simulation scenario is depicted in Table 5. In the first scenario, 2,000 malicious interest packets are bombarded by one compromised consumer. A total of 1,000 Normal Interest Packets were also induced in the system by a legitimate consumer in the same domain. In this scenario, no threshold value is set. The maximum throughput of a particular face is 100 bps. Initially, the throughput of the normal interest packets was up to 90% of the total capacity which is the desired result. But in the subsequent seconds, the Malicious packets entered the router. Queue capacity started to saturate, the throughput of the normal interest packet will start to drop as the queue gets filled up with the bombarded malicious packets. The processing overhead also started to increase because of the cost of the cache-miss penalty and verification overhead. This scenario is depicted in Fig. 11. In this second scenario, again 2,000 malicious interest packets are bombarded by one compromised consumer. A total of 1,000 Normal Interest Packets were also induced in the system by a legitimate consumer in the same domain. In this scenario, our proposed solution is placed and activated inside the NDN Router service manager. The maximum throughput of a particular face is 100 bps. The throughput of the normal interest packets was up to 90% of the total capacity which is the desired result. But in the subsequent seconds, the Malicious packets entered the router. Queue capacity started to saturate, then the proposed solution gets activated and blocks the malicious face when the cache miss counter value reached the threshold value. Then we can see that according to our simulation environment after 3rd-second malicious packets didn't enter the router queue and throughput of the normal interest packet will start to raise and other factors like processing overhead and queue capacity ratio get into the normal working range. This scenario is depicted in Fig. 12. A total of 2,000 Malicious Packets bombarded were detected and dropped successfully by our system. System accuracy proved to be 100%. Also, 1,000 legitimate special Interest packets were processed and no packet was dropped.

Comparison of throughput, queue capacity and processing overhead during the CPA special interest packet flooding attack and that of our proposed approach is summarized in Table 6.
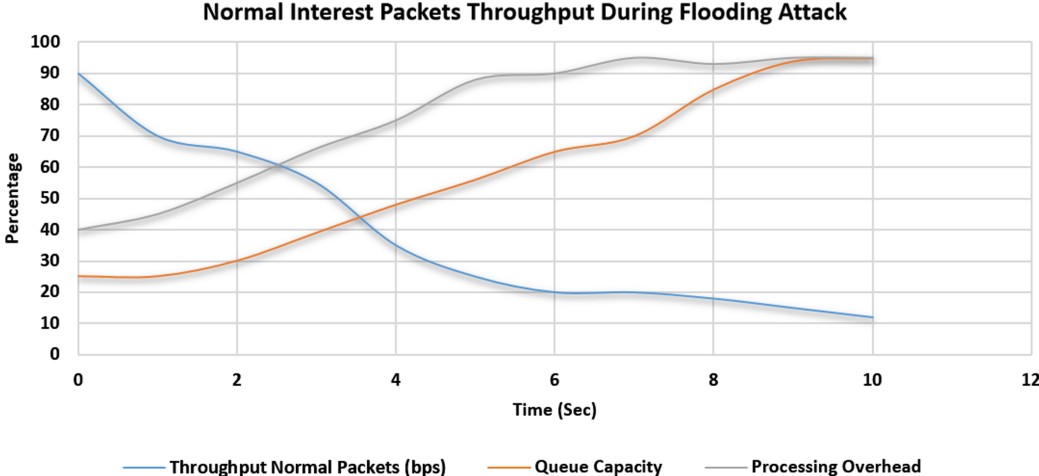

**Figure 11 Throughput of the normal special interest packets in flooding attack scenario.**

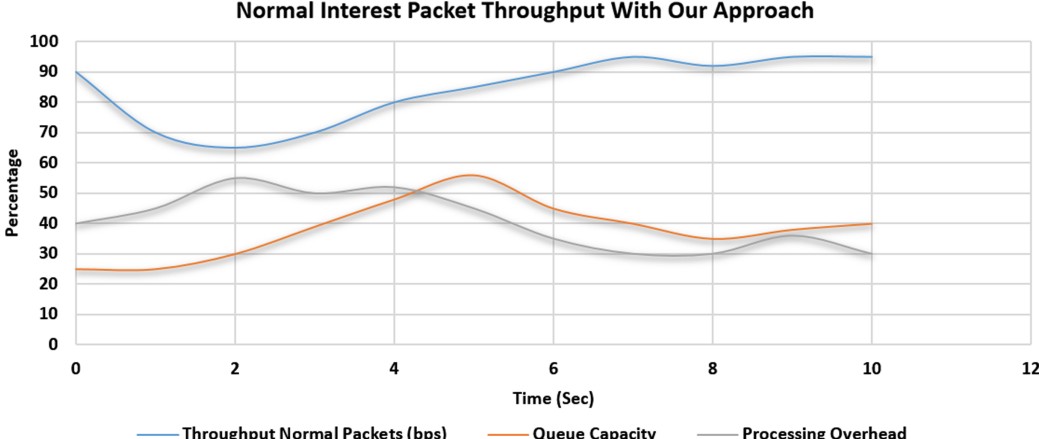

**Figure 12 Throughput of the normal special interest packets in flooding attack scenario with proposed mitigation strategy.**

**Table 6 Comparison of throughput, queue capacity and processing overhead during special interest packet flooding attack.**

| Category | Proposed approach | *DiBenedetto & Papadopoulos (2016)* |
|---|---|---|
| Min throughput of normal interest packet | 65% | 12% |
| Max queue occupation | 55% | 95% |
| Reporting packet size | Lightweight (Sha256 Hash - 32 Bytes) | Heavyweight (complete packet) |
| Trust anchor | Yes | Yes |
| Max processing overhead | 53% | 93% |
| Compromised consumer detection | Yes | Yes |
| Bogus report packet detection | Yes | Partial |

Effectiveness and efficacy of proposed solution by comparing the throughput of interest packets and queue capacity is elaborated in experiments as in Figs. 11 and 12 which is summarized in Table 6. It is evident from the experiments that during the Special Interest Packet flooding attack, our proposed approach showed promising results in terms of throughput, queue capacity and processing overhead.

## CONCLUSION AND FUTURE DIRECTION

The main contribution of this work is to devise a mechanism that identifies and prevents the compromised consumers from flooding the network with special Interest packets that are generated during the mitigation process of the Content Poisoning Attack. The compromised consumers place the hash of an un-poisoned content in the excluded filter field of the special interest packet which causes cache miss at the edge router. Owing to the bombardment of these special Interest packets, it will tremendously increase the processing overhead on the NDN Router. The cost is in terms of Cache-Miss penalty and verification overhead. Also, the queue capacity of the NDN Router gets saturated. Consequently, the legitimate requests from the other consumers get dropped or face a substantial amount of delays. We also observed the damaging effect of multiple malicious consumers flooding the edge router which was also well handled by using the proposed technique. After the implementation of our scheme in the Network Service manager at the NDN Edge Router, the malicious face will be blocked when the cache-miss ratio value reaches the specified threshold value. We also have made the threshold value dynamic by adjusting the initial threshold according to cache-miss ratio and queue capacity values. An improvement in this technique can be done by incorporating Quality of Service solutions in NDN Routers. Multiple Virtual queues for special Interest packets can be maintained in NDN Routers to handle the flooding of these packets. Different queuing disciplines and algorithms like Adaptive Virtual Queue (AVQ), Credit-Based Fair Queuing, Weighted Fair Queuing, Quick Fair Queueing, and Class-Based Queuing can be tested to augment our approach. Also, traffic shaping and rate control mechanism can be used to hold back the malicious face.

### Funding
The authors received no funding for this work.

### Competing Interests
Masood Ur-Rehman is an Academic Editor for PeerJ.

### Author Contributions
- Adnan Mahmood Qureshi conceived and designed the experiments, performed the experiments, analyzed the data, performed the computation work, prepared figures and/or tables, authored or reviewed drafts of the paper, and approved the final draft.

- Nadeem Anjum conceived and designed the experiments, analyzed the data, performed the computation work, prepared figures and/or tables, authored or reviewed drafts of the paper, and approved the final draft.
- Rao Naveed Bin Rais conceived and designed the experiments, performed the computation work, authored or reviewed drafts of the paper, and approved the final draft.
- Masood Ur-Rehman performed the experiments, prepared figures and/or tables, authored or reviewed drafts of the paper, and approved the final draft.
- Amir Qayyum analyzed the data, authored or reviewed drafts of the paper, provided the logistics to perform experiments, and approved the final draft.

### Data Availability

Code is available at Figshare:

Qureshi, Adnan (2021): MyNDNSim.zip. figshare. Software. DOI 10.6084/m9.figshare.13587650.v1.

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
