# Peer review of "Detection of malicious consumer interest packet with dynamic threshold values"

_PeerJ Computer Science, doi:10.7717/peerj-cs.435_

## Round 0.1 · original submission · Major Revisions

I have received reviews of your manuscript from two scholars who are experts on the cited topic. They find the topic very interesting; however, several concerns must be addressed regarding experimental results, threshold setting, comparisons with other approaches, types of attacks, vulnerabilities, and up-to-date references. These issues require a major revision. Please refer to the reviewers’ comments listed at the end of this letter, and you will see that they are advising that you revise your manuscript. If you are prepared to undertake the work required, I would be pleased to reconsider my decision. Please submit a list of changes or a rebuttal against each point that is being raised when you submit your revised manuscript.

Reviewer 1 ·

Basic reporting

abstract needs to be revised.
References are not sufficient
Figure 6. Mitigation of Flooding Attack-There is some ambiguity in the figure
The article should include sufficient introduction and background to demonstrate how the work fits into the broader field of knowledge. Relevant prior literature should be appropriately referenced.

Experimental design

The investigation must have been conducted rigorously and to a high technical standard.
The proposed method should be well discussed.
comparison with the existing approaches need to be highlighted

Validity of the findings

conclusions need to be revised -
Using the
358 mitigation technique mentioned in this paper, the Network Service Manager at NDN Edge router can
359 enable a mechanism in which upon reaching a certain threshold value; it blocks that interface temporarily
360 from which the excluded interest packets are being generated---Revise this statement

Additional comments

There are many state-of-the-art mitigation strategies for the content poisoning attack, but some new attack-surfaces have emerged with these schemes’ advent.
Mention what are some attacks.
Clearly mentions what is the outcome of the proposed method
Revise the abstract
Making the threshold value dynamic, which was initially set by the network operator statically? But in many networks, it is set dynamically -Comment
CPA Detection and Mitigation Approaches-need more discussion
Comparisons of CPA Mitigation Approaches- give the summary in a tabular form. Readers will easily understand
In Figure 3. Detection of Flooding Attack during Content Poisoning Attack Mitigation-interest packet generated and after again there is a block Interest Packet or Excluded Interest Packet
Discard Interest Packet with Exclude Filter and Exit-what is the role of this block
There is no clear discussion on EXPERIMENTAL RESULTS –
How the proposed method is superior to other methods
There is no comparison with other existing approaches. When there is no good comparison how can authors claim the method works fine?
There is no discussion on vulnerabilities
What are the various attack surfaces and how can they be resolved.
Discuss various types of attacks
Clearly mention the difference between the manual threshold and dynamic threshold. Give in a tabular for how can the proposed methods overcome the drawbacks of the manual method.
The threshold value is kept as 3. What does it mean by 3 and how can it be restricted.
The number of References needs to be increased

Reviewer 2 ·

Basic reporting

Authors' main contribution is to provide an additional security solution to face the content poisoning attack (CPA) which will prevent malicious consumers from flooding the network with unwanted reporting packet in the context of named data networking

Experimental design

Overall, the idea is clear and simple. Yet, the threshold adaptativity should be related to many oher metrics like the cache size and the history of the requested data (can adopt a trust establishment solution for instance).

Validity of the findings

My main concern regarding the authors findings is related to the validation environment. All NDN research community agreed on NDNSim over NS3 because it has all NDN modules already implemented. Why you did not use it?!

Additional comments

Figures quality should be improved. Please also include this work in the comparative table, so that we can see its position compared to what already exists.

I also recommend citing the following research works:
https://doi.org/10.1109/ICCCN.2019.8847146
https://doi.org/10.1016/j.future.2020.01.043

---

## Round 0.2 · Minor Revisions

The abstract still requires revisions. Please check English grammar, typo mistakes, and spelling. It is also of paramount importance to check the experimental results section, specifically Table 5. These issues require a minor revision. If you are prepared to undertake the work required, I would be pleased to reconsider my decision. Please submit a list of changes or a rebuttal against each point that is being raised when you submit your revised manuscript.

Reviewer 1 ·

Basic reporting

The abstract needs to be revised. what is the takeaway from the paper that should clearly be mentioned?
This article’s main contribution is the addition of a security mechanism in the CPA mitigation. This statement is not sufficient. Need to support with data.
check grammar and typo mistakes in the paper.
mention various types of attacks in tabular form

Experimental design

Table 5. Qualitative Comparison-- check the column Proposed Approach better support with values instead of mentioning Very High , high

Validity of the findings

Table 5. Qualitative Comparison -give reference numbers
This table should support with data and the comparisons should be in graphs i.e pictorial representation

Additional comments

check the paper against the grammatical and typo mistakes
visibility of the figures need to be enhanced
check the numbering of tables and figures

---

## Round 0.3 · accepted · Accept

I am pleased to inform you that your work has now been accepted for publication in PeerJ Computer Science.

Thank you for submitting your work to this journal.